# Spatiotemporal Analysis of Stranded Loggerhead Sea Turtles on the Croatian Adriatic Coast

**DOI:** 10.3390/ani14050703

**Published:** 2024-02-23

**Authors:** Željko Mihaljević, Šimun Naletilić, Jasna Jeremić, Iva Kilvain, Tina Belaj, Tibor Andreanszky

**Affiliations:** 1Croatian Veterinary Institute, Savska Cesta 143, 10000 Zagreb, Croatia; miha@veinst.hr (Ž.M.); andreanszky.vzr@veinst.hr (T.A.); 2Ministry of Environment and Energy, Radnička Cesta 80/7, 10000 Zagreb, Croatia; jasna.jeremic@mingor.hr; 3Blue World Institute of Marine Research and Conservation, Kaštel 24, 51551 Veli Lošinj, Croatia; belaj.tina@gmail.com

**Keywords:** loggerhead turtle, east Adriatic Sea, spatiotemporal analysis, Croatia

## Abstract

**Simple Summary:**

The Adriatic Sea is a crucial area for sea turtles, with identified hotspots of presence and areas of high risk. This research identifies distinct patterns in seasonal strandings, indicating season-specific abundance and age-specific mortality, particularly in the crucial neritic habitat of the northern Adriatic, whilst emphasizing key areas in Istria, Lošinj, Zadar, and Šibenik. High-density stranding occurrences, linked to factors like population density and tourism, have been observed in these regions. Fishing-induced mortality, collision with vessels, and potential cold stunning emerge as major threats to loggerhead turtles. Post-mortem investigations underscore the age-specific significance of longline fishing gear and vessel collisions in contributing to mortality, emphasizing the need for focused conservation efforts in high-risk areas. Recognizing and addressing mortality hotspots is essential for understanding and mitigating human–turtle interactions in the Adriatic.

**Abstract:**

This study investigates the spatiotemporal trends of loggerhead turtles along the Croatian Adriatic coast by using stranding data and post-mortem analyses. Information on 620 loggerhead turtles (*Caretta caretta*), collected in the period between 2010 and 2022, has been analysed. Seasonal stranding variations reveal distinct patterns, indicating season-specific abundance and age-specific mortality in different areas, particularly in the key neritic habitat of the northern Adriatic. The analysis identifies four critical areas in the northeast and central Adriatic showing high stranding densities and provides regional managers with a tool with which to effectively conserve and manage this species. Fishing-induced mortality, collision with vessels, and potential cold stunning are identified as major threats to loggerhead turtles. Post-mortem investigations reveal that longline fishing gear and collisions with vessels are significant age-specific mortality contributors, underscoring the need for targeted conservation efforts in high-risk areas. The study acknowledges potential biases in strandings records but highlights the importance of post-mortem investigations in understanding mortality causes. The findings provide valuable insights for improving conservation strategies, emphasizing the importance of focused surveillance and conservation efforts in identified high-risk locations to mitigate human–turtle interactions.

## 1. Introduction

The Adriatic Sea, a narrow semi-enclosed basin in the Central Mediterranean, is a rich neritic foraging habitat for sea turtles. Outside the breeding season, it is visited by juvenile and adult individuals of three distinct species of Adriatic sea turtles: the loggerhead (*Caretta caretta* Linnaeus, 1758), the green (*Chelonia mydas* Linnaeus, 1758), and the leatherback (*Dermochelys coriacea* Vandelli, 1761) sea turtle [1,2,3,4,5,6]. The northern Adriatic, a shallow basin with a maximum depth of about 100 m (average depth = 35 m), is a neritic area which serves as a habitat for both juvenile and adult loggerhead sea turtles, the most common sea turtle species in the Mediterranean Sea [1,4,7], whereas the southern Adriatic is an important oceanic developmental habitat for juvenile sea turtles, most of them arriving from Greek rookeries [3,8,9,10,11]. The central Adriatic is characterized by three depressions located along the transversal line off the coast of Pescara (Jabuka/Pomo Pit), with a maximum depth of about 280 m [12]. The eastern side is exposed to the fierce northeasterly wind known as bora, which descends from the Dinaric Alps and blows along the eastern Adriatic shoreline. The airflow can travel at high speeds towards the Italian coast, crossing enormous distances, and can rapidly decrease sea temperature [13]. Migratory routes in the Adriatic head northwards, along the eastern coast, following the current that enters the Adriatic [1,4,5,14,15,16,17,18,19,20,21,22,23,24,25]. The biggest threats to sea turtles are habitat degradation, accidental captures during fishing operations, intentional harvesting and consumption, egg exploitation, pollution, and vessel collisions [1,2,5]. The Adriatic has the highest incidental sea turtle catch rates in the whole Mediterranean across various kinds of fishing gear [6,26,27,28,29,30]. In order to breathe, sea turtles must surface or stay close to the surface. They may also stay at the surface for an extended period of time to rewarm and reoxygenate, which potentially exposes them to extensive marine traffic along migration routes [31,32,33,34]. Furthermore, sea pollution caused by plastic and chemical contaminants is undermining conservation efforts [35,36,37,38,39]. Thus, determining how loggerhead turtles use the Mediterranean Sea both spatially and temporally, and identifying areas of high risk, i.e., where their presence overlaps with human pressures, may be essential in order to establish effective conservation-related management measures [2,8,30,40,41,42].

Stranding records can be utilized to obtain data on spatiotemporal distribution, as well as other details about sea turtle populations, such as life phases, sex ratio, migratory habits, diet, and habitat [23,27,40,41,43,44,45]. It is important to understand that data obtained from turtle strandings could be biased. A number of variables, which vary greatly depending on the location and season, affect the likelihood that a dead or distressed sea turtle will drift ashore and be reported. These include carcass buoyancy, decomposition and scavenging rates, oceanographic and atmospheric conditions, shoreline characteristics, and detection probability [27,41,44,45]. The number of recorded strandings may only be a proxy for overall mortality and, more significantly, may not accurately reflect incidents, especially those in oceanic areas [43]. It is important to acknowledge that a sick turtle has a higher chance of being hit by a boat or interacting with fishing gear due to potential changes in their behavior. Furthermore, a floating turtle that is already dead has a higher chance of being hit by a boat, causing injuries that suggest a boat strike.

A comprehensive spatial–temporal analysis of sea turtles reported stranded in Croatia was carried out as part of this study in order to identify hazards and areas of high risk for sea turtles on the Croatian shoreline. Acquired data were used to establish protective measures for this highly endangered population.

## 2. Materials and Methods

For the purposes of this study, any sea turtle found alive or dead on the beaches or floating in coastal waters was defined as stranded. 

Data were collected on each stranded turtle encountered following standardized protocols [46,47]. When possible, data recorded included the following: species; location, date, and time of finding; size (midline curved carapace length (CCL in cm) and curved carapace width (CCW in cm); weight; sex; presence of epibionts; tag number; health status; decomposition status; therapy and care during rehabilitation; and causes of death [46,47]. The passive surveillance data presented in this study were collected within the area of the Croatian Adriatic Sea during a 13-year period, spanning from January 2010 to December 2022, as part of the protocol for the alerting and monitoring of dead, sick, or injured strictly protected marine species (marine mammals, sea turtles, and cartilaginous fish) mandated by the Croatian Ministry of Environment and Energy. When possible, the injured turtles received veterinarian help in the Sea Turtle Rescue Centers in the cities of Pula and Mali Lošinj.

The size class of stranded animals was estimated by using the CCL measures as reviewed in Casale (2018) [5]; that is, all sea turtles with CCL less than 40 cm were considered as small juvenile, those with CCL between 40 and 65 cm as large juvenile, and those with CCL greater than 65 cm as adult animals [2]. Turtles larger than 40 cm may swim independently of currents, so we used this CCL length as a provisional cut-off between small and large juvenile loggerhead turtles. The sex of the stranded animal was visually determined on site based on the size and muscularity of its tail. A systematic pathological examination was conducted on 60 loggerhead turtle carcasses submitted to the Laboratory of Pathology at the Croatian Veterinary Institute. Necropsy findings were noted in a standard turtle necropsy datasheet [47].

The stranding area was divided into three parts of the Adriatic Sea: northern, central, and southern. The northern Adriatic, extending between Venice and Trieste towards a line connecting Ancona and Zadar, is only 15 m deep at its northwestern end, and it gradually deepens towards the southeast, reaching a maximum depth of about 100 m (average depth = 35 m). The central Adriatic is located south of the Ancona–Zadar line, includes the 270-m deep Jabuka Pit, and extends to the 170-m deep Palagruža Sill, which separates the central from the southern Adriatic [12]. Surface sea temperature data were kindly obtained from the Croatian Meteorological and Hydrological Service. 

The spatiotemporal analysis of the seasonal distribution of stranded sea turtles was performed using the ANOVA statistical method with post hoc comparisons made using the Bonferroni correction. When ANOVA assumptions were not met, non-parametric Spearman correlation was used after testing for data normality and homoscedasticity via the Shapiro–Wilk test. Statistical analyses were performed using Stata 10.0 (StataCopr. 2016. Stata Statistical Software: Release 13.1, College Station, TX, USA). A level of *p* < 0.05 was considered significant, and, to indicate precision of our observations, a 95% confidence interval (CI) was presented. In order to understand the spatial distribution of stranded turtles in Croatia, we used the kernel density function. This function is based on the quadratic kernel function described by Silverman [48]. The analysis was performed in the ESRI@ArcGIS Desktop: Release 10.8 software, using Arc Tool Box > Spatial Analyst Tools > Kernel Density.

## 3. Results

A total of 620 loggerhead turtles were recorded in the database between January 2010 and December 2022 as part of the protocol mandated by the Croatian Ministry of Environment and Energy (Figure 1).

The tag numbers was recovered from 59 (9.51%) stranded animals. Size was measured in 379 (61.13%) stranded animals. Sixty (13.48%) dead turtles were necropsied to determine cause of death (Table 1).

An average of 47.69 (CI 42.5–52.89) individuals per year were recorded along the Croatian Adriatic coast. An above-average increase in the number of stranded turtles was recorded between 2013 and 2017, when 324 cases (52.25% of all strandings) were recorded in a span of five years. Another increase was recorded in 2021. The highest number of stranded cases was observed in the northern (*n* = 285, 45.97%), followed by the central (*n* = 207, 33.39%) and southern Adriatic (*n* = 128, 20.65%). Strandings were recorded mostly during summer (June, July, and August, *n* = 205, 33.06%) and winter (December, January, and February, *n* = 176, 28.39%). The lowest numbers were observed in March and September (*n* = 34, 5.48%).

The spatiotemporal distribution of stranding records is presented in Figure 2, and statistical analysis were evident in statistically significant differences in monthly records between sections (*p* = 0.001). Post hoc comparisons using the Bonferroni correction of monthly records by section showed significant differences between the northern and both the central and southern sections (*p* < 0.01). The differences found in the monthly records between the central and southern areas were not significant (*p* = 0.256). Most strandings in the northern Adriatic were recorded during June, July, and August (*n* = 118, 19.03%), when surface sea temperatures rise above 20 °C. In the central Adriatic, most cases were noted during January (*n* = 28, 13.53%) when surface sea temperatures fell below 11 °C, and in August (*n* = 25, 12.08%) when surface sea temperatures reached their peak. The majority of cases in the southern Adriatic were recorded during January (*n* = 24, 18.75%) and February (*n* = 17, 13.28%), coinciding with surface sea temperatures below 13 °C (Figure 2). In the northeast Adriatic, the kernel density analysis identified two geographically divided, high-density stranding areas during summer months, one along the Istrian peninsula and the other along the coast of Lošinj Island (Figure 3, Points 1 and 2). Two additional identified high-density stranding areas in the Zadar and Šibenik region are located in the central Adriatic (Figure 3, points 3 and 4).

The average size of stranded turtles was 56.53 cm (CI 54.06–58.99, range 10–180 cm). Monthly stranding records show no significant differences in relation to the size classes of turtles (*p* > 0.05). The number of strandings of small juvenile turtles with CCL < 40 cm was significantly different among stranding areas (*p* = 0.001). Such significant differences were not observed in larger size classes. (*p* > 0.05) (Table 2).

Sex was recorded in 174 turtles (28.06%), 66.67% of which were male and 33.33% female. If we consider only necropsy data, the sex ratio is female-skewed, with 41.67% male (*n* = 25; CI = 29.07–55.12%) and 58.34% female (*n* = 35; CI = 44.88–70.93%) based on necropsied specimens.

During the study, 60 individuals were submitted to a necropsy, 51 (CI = 73.43–92.90%) of which were found dead, and 9 (CI = 7.1–26.57%) died during treatment after having been found injured. The selection of individuals for necropsy was made based on the decomposition status of the carcasses. All carcasses submitted to the autopsy were fresh. Necropsies were performed on 17 animals (28.33%) from the northern section, 21 animals (35%) from the central section, and 22 animals (36.67%) from the southern section. The following results are from necropsies only (Table 3).

Interaction with fishing gear could be identified in 40% (*n* = 24; CI = 27.56–53.46%) of the carcasses. Of turtles that interacted with fishing gear, 62.5% (*n* = 15) were found to have interacted with longline gear, whereas 37.5% (*n* = 9) were found to have interacted with a fishing net. Specific pathological findings which indicated interaction with longline gear included drowning, a deeply placed fishing hook, and/or monofilament lines through the gastrointestinal tract with consequent fibrinonecrotic enteritis (Figure 4A). In total, longline fishing was found to be the cause of death in 25% (CI = 14.71–37.86%) of turtles, and 66.66% of them were juvenile animals (*n* = 10; CI = 38.38–88.18%). Autopsy findings of drowning, including the presence of froth or seawater in the bronchi and trachea, excessive fluid accumulations in small airways, and exudative or interstitial pneumonia, strongly suggested that entanglement in trawl gear was the cause of death in 15% (*n* = 9; CI = 7.1–26.57%) of animals. 

Trauma caused by vessel collisions was diagnosed in 33.33% (*n* = 20; CI = 21.69–46.69%) of animals, and 55% of them were adult animals (*n* = 11; CI = 31.53–76.94%). Pathological findings included extensive sharp-edged traumatic lesions of the head, neck, carapace, or plastron with corresponding trauma and hemorrhages in deeper tissues (Figure 4B). Treated animals exhibited trauma-related lesions/sepsis, subcutaneous oedema, fibrinous pleuritis and interstitial bronchopneumonia, fibrinous pericarditis, necrotizing splenitis, and interstitial nephritis. Vessel collisions were a major cause of death in adult animals, accounting for 55.0% (*n* = 11; CI = 31.53–76.94%), and in larger juvenile (CCL = 41–65 cm) animals, accounting for 28.57% (*n* = 6; CI = 11.28–52.17%) of deaths (Table 3).

Drowning after cold stunning was determined in eight (13.33%; CI = 5.94–24.59%) animals, 62.5% of which were subadult animals (*n* = 5; CI = 24.49–91.48%). It was diagnosed only in animals found ashore after a rapid decrease in sea temperature caused by the bora wind. The turtles were diagnosed with cachexia and showed minimal pathologic signs that included froth in the airways, cyanosis, hemorrhage of the skin on the neck and flippers, or interstitial pneumonia (present in treated animals (Figure 4C)).

Interstitial pneumonia was diagnosed in three (5%) animals, which died in rescue centers after treatment (Figure 4D). After unsuccessful treatment of extensive trauma, sepsis was diagnosed in six juvenile individuals (10%; CI = 3.76–20.51%). The differences between sections in conducted versus non-conducted turtle necropsies were notably significant (*p* = 0.004). Only 17 (5.96%) stranded animals from the northern section, 21 (10.14%) from the central section, and 22 (17.19%) from the southern section were subjected to necropsy. Consequently, necropsy findings cannot indicate potential differences in the cause of mortality between sections.

Longline fishing gear was found to be a major cause of death in small juvenile turtles (CCL mean size = 41.4 cm; CI = 29.11–53.69 cm; range = 20–100 cm), while colliding with a vessel was the prevailing cause of death in larger juvenile and adult turtles (CCL mean size = 64.11 cm; CI = 54.42–73.81 cm; range = 35–100 cm). Trawl nets (CCL mean size = 58.33 cm; CI = 41.32–75.35 cm; range 30–100 cm) and cold stunning affect a wide range of loggerhead turtle sizes (CCL mean size = 55.75 cm; CI = 41.86–69.64 cm; range = 18–70 cm). Due to small sample size, differences found in the determined cause of death among size class categories were not sufficiently conclusive to make any inference.

Irregular fragments of non-transparent, black, plastic caps with a range from 1 to 3 cm were found in the gastrointestinal tracts of two subadult animals, causing ulceration and hemorrhagic lesions with partial obstruction of the intestines. Hard fecal matter (fecaloma) composed of the shell of the Mediterranean mussel *Mytilus galloprovincialis* was found within the large bowel of 24 (40%; CI = 27.56–53.46) animals. Nematode Sulcascaris sulcata was detected in the gastrointestinal tract of two animals.

## 4. Discussion

Our study assessed the spatiotemporal trends of loggerhead turtles based on the data derived from an existing surveillance tool: stranded animals and post-mortem analysis of fresh carcasses. The analysis demonstrated the clear seasonality of strandings along the Croatian Adriatic coast, which is indicative of season-specific abundance in different areas and age-specific mortality related to region, which was observed among small juveniles.

Stranding seasonality occurs as a result of variation in the abundance, distribution, or mortality of animals, as well as the non-biological components of the stranding process, including environmental and climatic conditions, as well as observer effort. 

During summer, especially during July, stranding numbers were apparent in the northern Adriatic, matching the high densities of loggerhead turtles that were predicted for that area with the help of abundance surveys and fine-scale model predictions [11,24,25,49,50,51]. The northern Adriatic is the key neritic habitat for loggerhead turtles in the Mediterranean and, as hatchings were observed on the beach in the Venetian Lagoon, is a potential new nesting rookery [4,10,15,50,52,53,54]. It is, in fact, the largest foraging area in the Mediterranean, frequented by turtles hatched in western Greece, Crete, western Turkey, and, as has recently been documented, Albania [7,9,18,21,51,55]. 

The shallow waters, transitional habitats, and rich benthic communities of the northern Adriatic provide both ideal developmental grounds for juvenile animals in the benthic phase and an overwintering habitat [5,31,32].

Our analysis of stranding records designates the northern Adriatic as a significant area for the stranding of small juvenile turtles (CCL < 40 cm). Such a finding could be attributed to the high density of sea turtles in this area [17,20], but this should be considered with caution as high records could be a result of increased detection possibility of small animals during the summer months rather than a result of increased mortality. 

The central and southern Adriatic is a highly variable habitat for loggerhead turtles, which is consistent with earlier tagging studies and dispersal models [17,20]. 

With the onset of declining water temperatures during fall, adult and juvenile loggerhead turtles that utilize the summer neritic habitat in northern temperate waters migrate south towards southern wintering sites in order to utilize a broader foraging area [22,41,56]. With the highest number of recorded cases during the 13-year period, January was identified as the most critical winter month in the central and southern Adriatic.

Stranding density analysis identified four key areas that could be crucial for the conservation of the loggerhead turtle. 

The results suggest that in the northeast Adriatic, high-density stranding areas along the Istrian peninsula and the coast of Lošinj Island might be significant during the summer months. Turtle rescue centers were established in both areas, one in Pula and the other in Mali Lošinj. The results for the two additional key areas, Zadar and Šibenik (Figure 3, points 3 and 4), located in the central Adriatic, indicated that the majority of strandings occurred during January and August. Cold air temperatures, combined with the strong bora wind, lead to a rapid decrease in sea surface temperatures. This effect is particularly pronounced in shallow coastal embayments in those areas and can result in cold stunning. Critical water temperature thresholds for developing cold stunning in outdoor holding tanks range between 9.0 and 13.0 °C, with death occurring in the range between 5.0 and 6.5 °C [57,58]. Deaths caused by cold stunning could be prevented if detected and if the animal is transported to a rehab facility. Our data strongly suggest that a rescue center should be established in either of those central Adriatic areas. Human activities are highly concentrated in all four areas, where nearly all known anthropogenic stressors to loggerhead turtles occur and overlap. 

Despite significant indicators arising from our analyses, it should be emphasized that there is uncertainty regarding the extent to which the recorded strandings are representative of the at-sea populations and their distribution. In our study, the majority of the strandings were recorded on beaches near towns and tourist centers, which could indicate a reporting bias towards easier detection near populated areas along the Croatian coast. This could also be a potential reason for the observed discrepancy in the number of stranded animals submitted to necropsy among different areas. 

Sex determination based on the size and muscularity of the tail can be a challenging process when it comes to sea turtles, especially in juvenile animals. Such determination should be restricted to turtles with CCL > 75 cm [59]. Our results emphasize the sex determination differences between a stranding report and a necropsy report. Necropsy data indicate a slightly female-skewed sex ratio (41.67% of male and 58.34% of female sea turtles), similar to other studies [5].

To reduce the uncertainty around the use of information about stranded individuals when drawing conclusions about the influence on increased anthropometric inferences, we performed post-mortem investigations on 13.48% of dead turtles. 

It should be noted that estimates derived from autopsies of relatively fresh carcasses are subject to bias. Fresh carcasses alone were not a reliable indicator of turtle mortality because turtles killed by a fishing net further offshore would likely not appear on land as being suitable for autopsy. Therefore, our results should be considered with caution. Nevertheless, it was determined post-mortem that 40% of recorded mortality was associated with interaction with fishing gear, suggesting that by-catch is a major cause of turtle strandings along the Croatian Adriatic coast. According to interviews with fishermen in the northeast Adriatic, the estimated annual catch ranged from 657 to 4038 turtles, with an average of 2.8 turtles per vessel [60]. This suggests that the Croatian fishing fleet has a high impact on sea turtle mortality. More recent projections show that the yearly by-catch of turtles per vessel ranges from 1.80 in the southern area of the northern Adriatic to 5.30 in the Po River delta [29]. Therefore, the estimate that trawler by-catch causes 15% of loggerhead turtle death cases is likely heavily underestimated in our study. 

Our study indicates that longline fishing gear is a dominant by-catch mortality cause, with small juvenile turtles significantly impacted by interactions with longline fisheries. Longline fishing gear is reported to capture more turtles than any other fishing gear, with an estimated annual catch exceeding 200,000 loggerhead turtles worldwide and a mortality rate of 17–42% [5,60]. Juvenile loggerhead turtles have been characterized as discard scavengers. This behavior could potentially make them prone to feeding on baits on stationary fishing gear [44]. Most turtles are captured alive [61,62,63]. Usually, fishermen will release hooked turtles, cutting the line as soon as they can identify them. Based on the depth of hook ingestion and the length of monofilament lines, the estimated mortality rate after the release of the turtle is more than 30% [63]. Turtles with hooks in the lower esophagus and deeper had a mortality rate of 65% [63]. 

Accounting for 33.33% of deaths, collision with vessels is the second major threat to loggerhead turtles in the Adriatic [23,27,29,34]. High tourist activities along the coast are likely to result in increased recreational boat traffic. Additionally, many passenger companies increase their operations during the spring and summer months. Sea turtles surface primarily for breathing and remain close to the surface during activities such as foraging, resting, and mating. After diving deep below the thermocline, they may also spend extended periods of time at the surface in order to oxygenate and warm. When a ship hull or a propeller strikes a sea turtle, the collision typically results in severe, life-threatening injuries, such as extensive carapace fractures or deep cuts on the head, flippers, and carapace [64]. Any wound that penetrates the carapace is defined as a catastrophic injury, since it generally results in a compromised coelomic cavity, leading to infection and, ultimately, death. Small slicing wounds on the margin of the carapace (less than 4 cm in length) are considered to be an exception. In this area, the carapace and bone extend beyond the edge of the body cavity and small wounds are less likely to penetrate the coelomic cavity [65]. It has been found that turtles are unable to avoid being struck by a vessel at speeds higher than 4 km/h [64,65].

Overall, the presented results provide valuable information that can be used to improve conservation efforts for the loggerhead sea turtle. Surveillance and conservation activities need to be focused on areas of high risk for sea turtle mortality. In order to assess potential causal mechanisms for negative human–turtle interactions with stranding data, it is necessary to identify at-sea locations (hotspots) where those interactions occur.

## 5. Conclusions

This study provides crucial insights into the spatiotemporal trends and threats faced by loggerhead turtles along the Croatian coast. Identification of four critical stranding areas in the northeast and central Adriatic allows for targeted conservation efforts. Fishing-induced mortality, vessel collisions, and potential cold stunning are recognized as major threats. Post-mortem investigations emphasize the substantial impact of longline fishing gear and vessel collisions with age-specific significance. The findings advocate for focused surveillance and conservation measures in high-risk areas to mitigate human–turtle interactions and ensure the welfare of loggerhead turtle populations. One of targeted conservation efforts we propose to authorities is the establishment of mobile treatment facilities that could comprise a quick response measure to find debilitated and injured sea turtles (and other endangered and protected species) and provide first aid and secure transport to rescue centers.

## Figures and Tables

**Figure 1 animals-14-00703-f001:**
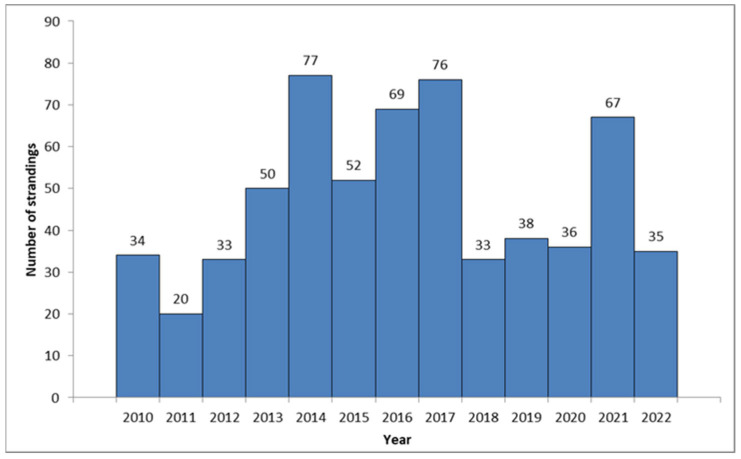
Annual records of observed *C. caretta* along the Croatian coast (2010–2023).

**Figure 2 animals-14-00703-f002:**
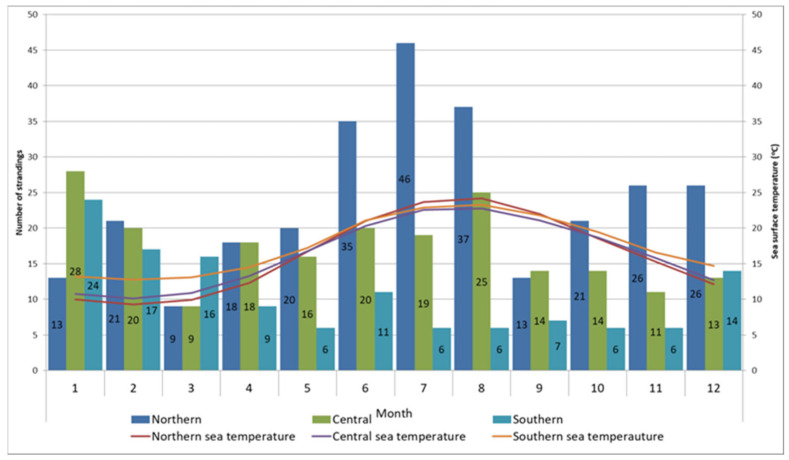
Monthly distribution of stranding records and sea surface temperature according to the stranding area. Numbers in column correspond to number of records within month and section.

**Figure 3 animals-14-00703-f003:**
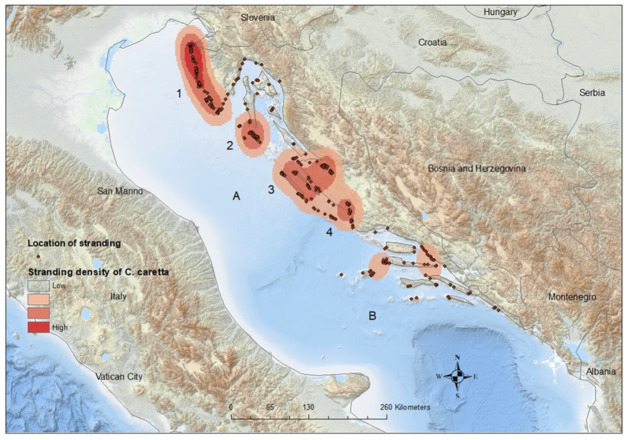
Spatial distribution of stranded loggerhead turtles along the Croatian Adriatic coast. Numbers correspond to identified high-density stranding areas: (1) Istrian peninsula; (2) Lošinj Island; (3) Zadar region; and (4) Šibenik region. Letters correspond to author’s division of stranding sections: (A) Ancona–Zadar line; (B) Palagruza still.

**Figure 4 animals-14-00703-f004:**
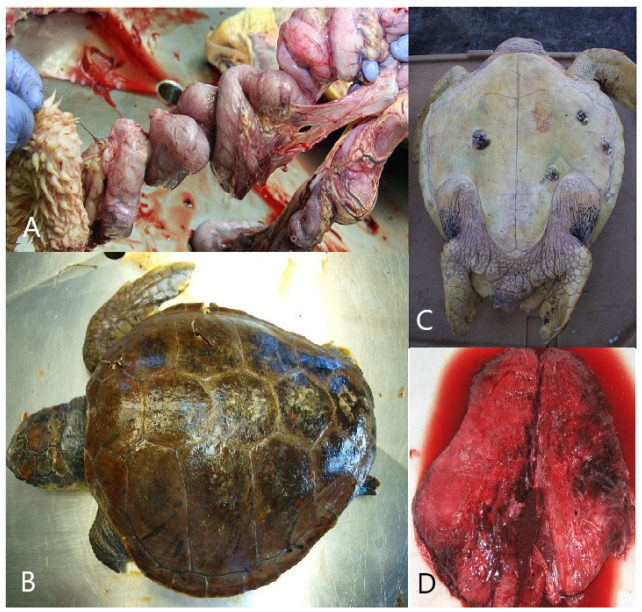
Gross appearance of interaction with longline gear, trauma caused by vessel collisions, drowning after cold stunning, and interstitial pneumonia. (**A**) Jejunum, stomach, and intestine: gross appearance of interaction with longline gear. (**B**) Dorsal side of the body: trauma of the carapace caused by vessel collisions. (**C**) ventral side of the body: cyanosis and hemorrhage of the skin on the neck and flipper. (**D**) Lungs: interstitial pneumonia.

**Table 1 animals-14-00703-t001:** Recorded state at finding of stranded turtle.

State of stranded animal		Number (percentage)
	Alive—unhurt	58 (9.37)
	Alive—hurt, treated, released	117 (18.87)
	Alive—treated, died	20 (3.23)
	Dead	425 (68.55)
	Total	620 (100)

**Table 2 animals-14-00703-t002:** Stranding location by size class of loggerhead turtles (percentages are shown in parentheses).

Stranding location		**CCL in cm**			
		**<40**	**41–65**	**>65**	**Total**
	Northern	70 (61.95)	61 (39.35)	44 (39.64)	175 (46.42)
	Central	24 (21.22)	61 (39.35)	41 (36.94)	126 (33.25)
	Southern	19 (16.81)	33 (21.29)	26 (23.42)	77 (20.58)
	Total	113 (100)	155 (100)	111 (100)	379

**Table 3 animals-14-00703-t003:** Determined cause of death by size class of loggerhead turtles.

Cause of death		**CCL in cm**			
		**<40**	**41–65**	**>65**	**Total**
	Fishing net	2 (10.53%)	3 (14.29%)	4 (20%)	9 (15%)
	Longline gear	10 (52.63%)	3 (14.29%)	2 (10%)	15 (25%)
	Vessel collision	3 (15.79%)	6 (28.57%)	11 (55%)	20 (33.33%)
	Cold stunning	1 (5.26%)	5 (23.51%)	2 (10%)	8 (13.33%)
	Interstitial pneumonia	1 (5.26%)	1 (4.76%)	1 (5%)	3 (5%)
	Sepsis	2 (10.53%)	3 (14.29%)	0	5 (8.33%)
	Total	19 (100%)	21 (100%)	20 (100%)	60 (100%)

## Data Availability

The data presented in this study are available on request from the first author.

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
