# Peer review of "Spatiotemporal Analysis of Stranded Loggerhead Sea Turtles on the Croatian Adriatic Coast"

_animals, 2024, doi:10.3390/ani14050703_

Round 1
Reviewer 1 Report
Comments and Suggestions for Authors
The paper is very well written. There a a few minor things to address but otherwise a good start.
Somewhere within the paper the authors need to better account for/talk about near shore bias and what that means for their conclusions.
Somewhere within the paper the authors need to better address the differences between all the records. Eg there were 425 animals reported dead however they only analyze 379 to carapace size. Also all their conclusions are based on necropsy of 60 animals (13.5% of dead animals). I understand that is part of using data which is already collected but several other authors have accounted for this. Try look at Epperly 1996, Flint 2015 or Read 2023.
It might also be nice to look at Cause of death by geographic location to see if different impacts occur in different areas. The authors have identified the areas where most strandings occur but what about by cause, this may impact what rehabilitation facilities focus on.
The authors need to ensure when they report CI and % they report N numbers too.

I have included them above and in the document
Author Response
Authors’ Response to Reviewer 1
Point 1. L90-93 Authors need to clarify this calculation
Response 1. The calculation has been corrected and the sentence rewritten.
L 94–99 “The passive surveillance data presented in this study was collected in the Croatian Adriatic Sea during a 13-year period, spanning from January 2010 to December 2022, as part of the Protocol for Alerting and Monitoring of dead, sick or injured strictly protected marine species (marine mammals, sea turtles, and cartilaginous fish) mandated by the Croatian Ministry of Environment and Energy.”
Point 2. L 105-106 A map be useful here
Response 2. The authors have decided to correct Figure 3 by presenting different colors and key points A and B in relation to the part of the text in which they divide the stranding area on the Croatian coastline.
Changes have been made to the text, Figure 3, and figure captions.
L 112–118 “The northern Adriatic, extending between Venice and Trieste towards a line connecting Ancona and Zadar (Figure 3, Point A), is only 15 m deep at its northwestern end, and it gradually deepens towards the southeast, reaching a maximum depth of about 100 m (average depth = 35 m). The central Adriatic is located south of the Ancona–Zadar line, includes the 270-m deep Jabuka Pit and extends to the 170-m deep Palagruža Sill (Figure 3, Point B), which separates the central from the southern Adriatic [12].”
*Numbers correspond to identified high-density stranding areas, 1 - Istrian peninsula, 2 - Lošinj Island, 3 - Zadar region, and 4 - Šibenik region. Letters correspond to authors’ division of the stranding sections, A - Ancona-Zadar line and B - Palagruža Sill.
Point 3. L131 suggest rehabilitation is a better word
Response 3. The paragraph has been deleted as suggested by another reviewer.
Point 4. L132 suggest rehabilitation is a better word
Response 4. The paragraph has been deleted as suggested by another reviewer.
Point 5 L133-134 What % of tag recoveries of the total tagged population is this? It might be meaningful.
Response 5. Authors could not find data on the total tagged population of sea turtles in the Mediterranean or Adriatic (http://www.seaturtle.org/tagfinder/). The Blue World Institute (author Tina Belaj) has been involved in many projects tracking turtles and they applied 5–10 tags on loggerhead turtles per year. The Croatian Ministry of Environment and Energy (author Jasna Jeremić) does not have this information either.
Point 6. L136-139 “The highest number of stranded turtles was recorded between 2013 and 2017, when 324 cases (52.25% of all strandings) were recorded in a span of five years. An average of 47.69 (CI 42.5–52.89) individuals per year were recorded along the Croatian Adriatic coast. “
This paragraph needs to be reworded to better highlight how high the strandings were otherwise it’s just saying that 50% of strandings occurred in half the study time
Response 6. This part has been rewritten.
L 144–146 “An above average increase in the number of stranded turtles was recorded between 2013 and 2017, when 324 cases (52.25% of all strandings) were recorded in a span of five years. Another increase was recorded in 2021.”
Point 7. L 174 Include a statement here to say the results following are from the necropsies only. eg confirmed vs unconfirmed.
Response 7. The statement has been included.
L 195 “The following results are from necropsies only (Table 3).”
Point 8. L 177 Add N
Response 8. This paragraph has been rewritten, as suggested by another reviewer.
Point 9. L 181 Add N
Response 9. This paragraph has been rewritten, as suggested by another reviewer.
Point 10. L 185 Add N
Response 10. This paragraph has been rewritten, as suggested by another reviewer.
Point 11. L 189 “Pathological findings included extensive sharp-edged traumatic lesions of the head, neck, carapace or plastron” This needs careful consideration. If an animal has prop cuts to its plastron, was the animal right side up at the time of impact or was it already floating upside down?
Response 11 Many thanks to the reviewer for pointing out a possible interpretation dilemma. The sentence has been rewritten to address trauma of live loggerhead turtle caused by a vessel collision.
L 211–214 “Pathological findings included extensive sharp-edged traumatic lesions of the head, neck, carapace or plastron with corresponding trauma and haemorrhages in deeper tissues (Figure 4, B).”
Authors’ comment: Postmortem trauma was differentiated by non-existing haemorrhage in adjacent tissue.
Point 12. L196 -197 “It was diagnosed in animals found ashore after a rapid decrease in sea temperature caused by the wind bora.” I not familiar with this but I think it should be bora win
Response 12 The sentence has been rewritten as suggested.
L 220–222 “It was diagnosed only in animals found ashore after a rapid decrease in sea temperature caused by the bora wind.”
Point 13. L205 Table 3. Determined cause of death by size class of loggerhead turtles
With this table think about whether it should be % per size class or % by cause. Both are meaningful. By cause could be more impactful for long line.
Response 13. After careful consideration, the authors have decided to leave % per size class because such a presentation allows for significant differences found in the determined cause of death among size class categories to be visualized.
Point 14. L 229-231: The northern Adriatic is the key neritic habitat for loggerhead turtles in the Mediterranean, and, as hatchings occur on the beach in the Venetian Lagoon, a potential new rooking area [4,10,15,50,52–54].
This sentence needs to be clarified/linked better. Is it a new nesting rookery?
Response 14. The sentence has been rewritten
L 268–270 “The northern Adriatic is a key neritic habitat for loggerhead turtles in the Mediterranean, and, as hatchings were observed on the beach in the Venetian Lagoon, is a potential new nesting rookery [4,10,15,50,52–54].”
Point 15 and 16. Lines 274-277 “Our results emphasize the sex determination differences between a stranding report and an autopsy report. Autopsy data indicate a slightly female-skewed sex ratio (41.67% of male and 58.34% of female sea turtles), similar to other studies [5].” suggest necropsy is a better word than autopsy
Response 15 and 16. L 316 Autopsy has been replaced by necropsy.

Reviewer 2 Report
Comments and Suggestions for Authors
Should abstracts be written in past tense?
L57- harvesting not killing
L62 potentially exposes
L80 was used
L85- defined as stranded
L105 was divided.
Tables 1 & 3. Repeat of data presented in paragraphs above. Remove text or table.
L184- All the clinical signs suggest are drowning. This may be due to forced submersion, but you cannot make that call with additional evidence. Submerged drowning uses shows as "dry drowning" where the epiglottis is locked and no water enters the lungs (the animal dies of asphyxiation).
L195-200: cold stunning in this case is the presumptive diagnosis. Be careful how present.
L212- list size of plastic caps- volume and size of plastic is important to determine contribution to cause of death.
Results; were there any parasites? Interesting to not have any infections of spirorchiids or coccidia. Worth noting if the case.
L235- must validate why this is a critical area.
L251-266: be careful selecting sites as this may be caused by drift from currents verses a common site for stranding. Assess based on live vs dead strandings to see if really "hotspots". You should consider mobile treatment facilities vs 4 bricks and mortar facilities. Can move around seasonally and respond to events.
L310 "and warmth" then delete the following sentence with refs.
Comments on the Quality of English Language
Check on tense and determine if journal wants it written in past tense or if present tense is ok, please.
Author Response
Authors’ Response to Reviewer 2
Point 1. Should abstracts be written in past tense?
Response 1. We have applied general rules regarding tense use:
Use present tense while stating general facts.
Use past tense when writing about prior research.
Use past tense when stating results or observations.
Use present tense when stating the conclusion or interpretations.
Use present tense when referring to your study/paper.
Point 2. L57- harvesting not killing
Response 2 L 57 The sentence has been rewritten as suggested.
Point 3. L62 - potentially exposes
Response 3 L 62 The sentence has been rewritten as suggested.
Point 4. L80 was used
Response 4. L 83 The sentence has been rewritten as suggested.
Point 5. L85- defined as stranded
Response 5. L 88 The sentence has been rewritten as suggested.
Point 6. L105 was divided.
Response 6. L 111 The sentence has been rewritten as suggested.
Point 7. Tables 1 & 3. Repeat of data presented in paragraphs above. Remove text or table.
Response 7. The authors have decided to change the text and leave Table 1.
“A total of 425 (68.50%) turtles were found dead, 20 (3.23%) were injured and later died during recovery, 117 (18.87%) were injured and either recovered or are currently in recovery, and 58 (9.35%) were found alive but unharmed.” – the sentence has been deleted.
The authors have decided to change the text and leave Table 3. Some sentences have been rewritten; additional information required by reviewer 1 has been added.
L 204–206 “In total, longline fishing was found to be the cause of death in 25% (CI 14.71–37.86%) of turtles, 66% of which were juvenile animals (N = 10, CI 38.38–88.18%).”
L 211–212 “Trauma caused by vessel collisions was diagnosed in 33.33% (N = 20, CI 21.69– 46.69%) of animals, 55% of which were adult animals (N=11, CI 31.53–76.94%).”
L 220–221 “Drowning after cold stunning was determined in eight (13.33%, CI 5.94–24.59%) animals, 62.5% of which were subadult animals (N = 5, CI 24.49–91.48%).”
“Longline gear caused 52.63% (N = 10, CI 28.86–75.56%) of deaths in small juvenile animals (Table 3).” – the sentence has been deleted.
Point 8. L184- All the clinical signs suggest are drowning. This may be due to forced submersion, but you cannot make that call with additional evidence. Submerged drowning uses shows as "dry drowning" where the epiglottis is locked and no water enters the lungs (the animal dies of asphyxiation).
Response 8. L181. Many thanks to the reviewer for pointing out a possible diagnostic problem with drowning. The authors have explained how drowning was diagnosed.
L 206–210 “Necropsy findings of drowning, including the presence of froth or seawater in the bronchi and trachea, excessive fluid accumulation in small airways, as well as exudative or interstitial pneumonia strongly suggested that entanglement in trawl gear was the cause of death in 15% (N = 9, CI 7.1–26.57%) of animals.”
The authors did not distinguish wet, dry or secondary drowning by performing a histopathology of the lung tissue. Histopathology was limited by autolytic changes on carcasses. Excessive fluid accumulation in the small airways strongly indicates wet drowning.
Point 9. L195-200: cold stunning in this case is the presumptive diagnosis. Be careful how present.
Response 9. Diagnostic uncertainty on drowning after cold stunning was reduced by anamnestic data on the rapid decrease of sea temperature by the bora wind. The sentence has been rewritten.
L 221–223 “It was diagnosed only in animals found ashore after a rapid decrease in sea temperature caused by the bora wind.”
Point 9. L212- list size of plastic caps- volume and size of plastic is important to determine contribution to cause of death.
Response 10. The sentence has been rewritten, required additional information has been added.
L 250–252 “Irregular fragments of non-transparent black plastic caps, ranging from 1 to 3 cm in size, were found in the gastrointestinal tracts of two subadult animals, causing ulceration and haemorrhagic lesions with partial obstruction of the intestines.”
Point 10. Results; were there any parasites? Interesting to not have any infections of spirorchiids or coccidia. Worth noting if the case.
Response 10. Additional information has been added
L 254 “Nematode Sulcascaris sulcata was detected in the gastrointestinal tract of two animals.”
Point 11. L237- must validate why this is a critical area.
Response 11. The sentence has been corrected.
“Our analysis of stranding records designates the northern Adriatic as a significant area for the stranding of small juvenile turtles (CCL < 40 cm).”
Point 12. L251-266: be careful selecting sites as this may be caused by drift from currents verses a common site for stranding. Assess based on live vs dead strandings to see if really "hotspots". You should consider mobile treatment facilities vs 4 bricks and mortar facilities. Can move around seasonally and respond to events.
Response 12. Stranding records have multiple biases that have been addressed in the manuscript (L309–313; 323). With the help of abundance surveys, it is possible to predict potential stranding areas in detail. Many thanks to the reviewer for sharing the idea about mobile treatment facilities. Mobile treatment facilities could quickly respond to found debilitated and injured sea turtles (and other endangered species), provide first aid and transport to rescue centres. Such mobile treatment facilities are feasible, involve citizen science and help sea turtle conservation.
With your permission the authors would like to add the following sentence to the Conclusion.
L 376–380 “One of the targeted conservation efforts we propose to authorities is the establishment of mobile treatment facilities that could be a quick response measure to found debilitated and injured sea turtles (and other endangered and protected species) and could provide first aid and secure transport to rescue centres. “
Point 13. L310 "and warmth" then delete the following sentence with refs.
Response 13. L 352 The sentence has been rewritten.

Reviewer 3 Report
Comments and Suggestions for Authors
See attached document

See attached document
Author Response
Authors’ Response to Reviewer 3
Point 1. In the results section values for CI are presented. Is this Confidence Intervals? Need to clearly state what CI is. If it is Confidence Intervals, it needs to be included and explained in the methods. If using CI, need to state what CI level is being used and what larger population is being referred to, all loggerheads in the Adriatic? Need to make sure CI is being appropriately used. Strandings are a biased sample of the larger population, as the Authors state, so not sure it is appropriate to calculate CI values.
Response 1. The authors have decided to note a 95% confidence interval (CI). The notion of a 95% confidence interval aids in evaluating estimate accuracy and facilitates determining the range of values that may encompass the actual parameter value in the population. Consequently, it serves as a crucial statistical instrument, offering contextual insights into the trustworthiness of estimates and supporting informed decision-making in research.
Changes have been made.
L 126–128 “A level of p < 0.05 was considered significant, and, to indicate the precision of our observations, a 95% confidence interval (CI) was presented.”
INTRODUCTION
Point 2. L 70 – 77 Authors mention possible bias working with stranding data, which is fantastic, but missed a very important bias when looking at the possibility of boat strike mortality. It is important to acknowledge that a floating turtle that is already dead or even sick has a higher chance of being hit by a boat causing injuries that suggest a boat strike as the cause of death rather than occurring after death. This have been considered by the authors, but it needs to be included in the bias section and clearly addressed in the paper.
Response 2. Many thanks to the reviewer for pointing out potential diagnostic bias of a boat strike as the cause of death. The text has been rewritten.
L 78–81 “It is important to acknowledge that a sick turtle has a higher chance of being hit by boat or interacting with fishing gear due to a potential change in behaviour. Furthermore, a floating turtle that is already dead has a higher chance of being hit by a boat causing injuries that suggest a boat strike.”
The authors acknowledge that it is very hard to identify trauma caused by a boat strike in a decomposed carcass, therefore, the sentence has been rewritten to address trauma of live loggerhead turtles caused by vessel collisions.
L 212–215 “Pathological findings included extensive sharp-edged traumatic lesions of the head, neck, carapace or plastron with corresponding trauma and haemorrhages in deeper tissues (Figure 4, B).”
MATERIALS AND METHODS
Point 3. Lines 86 – 87 Sentence is awkward. Maybe “Data was collected on each stranded turtle encountered following standardized protocols [46,47]. When possible, data recorded included: species, location …”
Response 3. The sentence has been rewritten as suggested.
L 90–94 “Data was collected on each stranded turtle encountered following standardized protocols [46,47]. When possible, data recorded included: species, location, date and time of finding, size (midline curved carapace length (CCL in cm) and curved carapace width (CCW in cm)), weight, sex, presence of epibionts, tag number, health status, decomposition status, therapy and care during rehabilitation, as well as causes of death.”
Point 4. Line 88 How was sex determined for stranded animals? Sex ratios based on stranding data and necropsies are referenced later as being in opposition, authors should add a sentence on how sex was determined in stranded animals.
Response 4. A sentence has been added to the text
L 107–108 “The sex of the stranded animal was visually determined on site based on the size and muscularity of its tail.”
Point 5. Line 90 Should be 13-year period, not 22
Response 5. The sentence has been corrected as suggested.
Point 6. Line 97 Check on how to cite Casale, 2018. Should this be a Number like all other citations? Or does a number need to be added to the citation. What is the full citation, is it already listed?
Response 6. The sentence has been corrected as suggested.
L 102 “The size class of stranded animals was estimated by using the CCL measures as reviewed in Casale [5].”
Point 7. Line 99 Change to read “CCL great than 65 cm as adult…” not 66 cm, based on Table 2. If this is not correct, how were turtles that measured 65 cm age classified?
Response 7. The sentence has been corrected as suggested.
L 103–105 “Namely, all sea turtles with CCL less than 40 cm were considered as small juvenile, those with CCL between 40 and 65 cm as large juvenile, and those with CCL greater than 65 cm as adult animals [2].”
Authors’ comment: Turtles measured 65 cm were classified as large juvenile specimens.
Point 8. Line 105 – 106 Throughout the paper the 3 parts of the Adriatic are referred to as northern, central and southern. But here, when defining the 3 parts, they are called north, central and south. Be consistent.
Response 8. The paper has been corrected as suggested.
L 112–113 “The stranding area was divided into three parts of the Adriatic Sea: northern, central, and southern.”
L 170 “Figure 2. Monthly distribution of stranding records and sea surface temperature according to the stranding section”
L184
Table 2. Stranding location by size class of loggerhead turtles
|
Stranding location |
|
CCL in cm |
|
||
|
|
< 40 |
41–65 |
> 65 |
Total |
|
|
Northern |
70 (61.95%) |
61 (39.35%) |
44 (39.64%) |
175 (46.42%) |
|
|
Central |
24 (21.22%) |
61 (39.35%) |
41 (36.94%) |
126 (33.25%) |
|
|
Southern |
19 (16.81%) |
33 (21.29%) |
26 (23.42%) |
77 (20.58%) |
|
|
Total |
113 (100%) |
155 (100%) |
111 (100%) |
379 |
|
Point 9. Lines 107 – 110 Use just ‘m’ and do not write out ‘metre’ Three places in this paragraph, and check rest of paper
Response 9. The text has been corrected as suggested.
L 113–195 “The northern Adriatic, extending between Venice and Trieste towards a line connecting Ancona and Zadar, is only 15 m deep at its northwestern end, and it gradually deepens towards the southeast, reaching a maximum depth of about 100 m (average depth = 35 m). The central Adriatic is located south of the Ancona–Zadar line, includes the 270-m deep Jabuka Pit and extends to the 170-m deep Palagruža Sill, which separates the central from the southern Adriatic [12].
RESULTS
Point 10. Figure 1 Why no Y-axis scale or label? No X-axis label
Response 10. Figure 1 has been corrected as suggested.
Point 11. Line 134 Cause of death is not listed in Table 1 and there is nothing in Table 1 that indicates the 60 turtlethat were necropsied. Just remove reference to Table 1. May want to change to “Sixty (13.48%) deadturtles were necropsied to determine cause of death.”
Response 11. The text has been changed as suggested.
L 141 “60 (13.48%) dead turtles were necropsied to determine the cause of death.”
Point 12. All of Table 1 data is already presented in the text. Don’t need it all in the text and all in a Table.
Response 12. The text has been deleted.
Point 13. Line 138 Generally Standard Error (SE) and range is provided when reporting an average. Need to explain in methods why CI is being reported and if justified, what CI level is being used.
Response 13. Same as response 1.
L 126–128 “A level of p < 0.05 was considered significant, and, to indicate the precision of our observations, a 95% confidence interval (CI) was presented.”
Point 14. Line 141 Why is CI being reported for the northern strandings and not the others? Not sure if you really need CI for these results, find to just state the percentage.
Response 14. The text has been changed as suggested.
L 147–149 “The highest number of stranded cases was observed in the northern (N = 285, 45.97%), followed by the central (N = 207, 33.39%) and southern Adriatic (N = 128, 20.65%).”
Point 15. Lines 145 – 149 Authors refer to the 3 parts as areas in this sentence. Be consistent. I would suggest referring to them as “sections”. Suggest changing to “showed statistically significant differences in monthly records among sections (p = 0.001).”
Response 15. Text has been changed as suggested.
L 153–155 “The spatiotemporal distribution of stranding records is presented in Figure 2, and statistical analysis shows statistically significant differences in monthly records between sections (p = 0.001).”
Point 16. The next two sentences seem to contradict each other: Post hoc comparisons using the Bonferroni correction of monthly records by area showed significant differences between the northern, central and southern Adriatic (p < 0.01). The differences found in the monthly records between the central and southern areas were not significant (p = 0.256).Was the significant difference between the northern section and both the central and southern sections?
Response 16. Authors’ comment: Yes, significant difference was observed between the northern section and both the central and southern sections. The text has been rewritten as suggested.
L155–157 “Post hoc comparisons using the Bonferroni correction of monthly records by section showed significant differences between the northern section and both the central and southern sections (p < 0.01).”
Point 17. Lines 149 – 154 Why no temperature data for the central and southern sections? No relationship between temperature and stranding numbers? For the southern section it appears temperature is related to water temp.
Response 17. The text has been changed as suggested.
L 160–165 “In the central Adriatic, most cases were noted in January (N = 28, 13.53%) when surface sea temperatures fell below 11°C, and in August (N = 25, 12.08%) when surface sea temperatures reached their peak. The majority of cases in the southern Adriatic were recorded during January (N = 24, 18.75%) and February (N = 17, 13.28%), coinciding with surface sea temperatures below 13°C (Figure 2).”
Point 18. Line 154 Text does not refer to Figure 3. Remove reference. Figure 3 really isn’t discussed until the Discussion, see comments later related to this.
Response 18. The text has been changed as suggested.
Point 19. Figure 2 Need to add label to Y-axis on both the left (sea surface temperature) and right (number of strandings). Figure title needs to include more mention of sea surface temperature.
Response 19. Figure 2 has been changed as suggested.
Point 20. Figure 3. Not really described in results text, what do the numbers represent? This could be referred to at Line 142 after the sentence about percentage of strandings in each section. Would be good to also add lines on map indicating northern, central and southern sections.
Use a color other than blue to represent stranding density, blue is also representing water depth and brown represents land
Response 20. Changes have been made to the text, Figure 3, and figure captions.
L 113–118 “The northern Adriatic, extending between Venice and Trieste towards a line connecting Ancona and Zadar (Figure 3, Point A), is only 15 meters deep at its northwestern end, and it gradually deepens towards the southeast, reaching a maximum depth of about 100 m (average depth = 35 m). The central Adriatic is located south of the Ancona–Zadar line, includes the 270-m deep Jabuka Pit and extends to the 170-m deep Palagruža Sill (Figure 3, Point B), which separates the central from the southern Adriatic [12].”
Figure 3. Spatial distribution of stranded loggerhead turtles along the Croatian Adriatic coast.
*Numbers correspond to identified high-density stranding areas, 1 - Istrian peninsula, 2 – Lošinj Island, 3 - Zadar region, and 4 - Šibenik region. Letters correspond to the authors’ division of stranding sections, A – Ancona–Zadar line and B – Palagruža Sill
Point 21. Table 2 Text about percentages should be placed in table Title. Place table after first reference on Line 167.
Response 21. Table 2 has been placed after the first reference; percentages have been added to the parentheses to be consistent with Table 3.
Point 22. Line 163 Please provide the range of sizes.
Response 22. The text has been corrected as suggested.
L 179 “The average size of stranded turtles was 56.53 cm (CI 54.06–58.99, range 10–180 cm)”
Point 23. Lines 163 – 165 Monthly stranding records show no significant differences in relation to the size classes of turtles (p = 0.208, p = 0.668, and p = 0.118, respectively). Respectively to what? Age class?
Response 23. Authors’ comment: P values were given for each size class, and the word respectively could be misleading. The text has been corrected as suggested.
L 180–182 “Monthly stranding records show no significant differences in relation to the size classes of turtles (p > 0.05).”
Point 24. Lines 165 – 167 unclear, please rewrite to clarify.
Response 24. The text has been rewritten.
L 181–183 “The number of strandings of small juvenile turtles with CCL < 40 cm was significantly different among stranding areas (p = 0.001). Such significant differences were not observed in larger size classes (p > 0.05) (Table 2).”
Point 25 . Line 168 Change to “Sex was determined in 174 stranded turtles…” reporting CI doesn’t seem appropriate here. If you report it anywhere, it would be for the percentage of males and females, like what was done for the necropsy results.
Response 25. The text has been corrected as suggested, CI has been deleted.
L 186 “Sex was recorded in 174 turtles (28.06%), 66.67% of which were male and 33.33% of which were female.”
Point 27. Lines 169 – 171 Change to read “In contrast, the sex ratio is female-skewed, with 41.67% male (N = 25, CI 29.07–55.12%) and 58.34% female (N = 35, CI 170 44.88–70.93%) based on necropsied specimens.
Response 27. The text has been corrected as suggested.
L 187–189 “In contrast, the sex ratio is female-skewed, with 41.67% male (N = 25, CI 29.07–55.12%) and 58.34% female (N = 35, CI 170 44.88–70.93%) animals, based on necropsied specimens.”
Point 28. How many of 174 stranded turtles where sex was determined were also necropsied? If sex was determined at stranding and the turtle was also necropsies, how many was the sex correctly identified?
Could the difference between stranding and necropsy sex identification be an issue of misidentification at stranding?
Response 28. Authors’ comment: Data on sixty necropsied turtles was reported to the Croatian Ministry of Environment and Energy and included in the records. Fresh carcasses were immediately transported to necropsy and all data were reported afterwards. The text has been corrected as suggested, and a new reference has been added.
L 314–316 “Sex determination based on the size and muscularity of the tail can be a challenging process when it comes to sea turtles, especially in juvenile animals. Such determination should be restricted to turtles with CCL > 75 cm [59].”
Casale, P.; Freggi, D.; Basso, R.; Argano, R. Size at male maturity, sexing methods and adult sex ratio in loggerhead turtles (Caretta caretta) from Italian waters investigated through tail measurements. Herpetol. J. 2005, 15, 145–148.
Point 29. Line 172 – How were the 60 turtles selected? How many turtles were necropsied from each part/section of the study area?
Response 29 Additional information has been added to the text.
L 192–196 “The selection of individuals for necropsy was made based on the decomposition status of the carcasses. All carcasses submitted to the necropsy were fresh. Necropsies were performed on 17 animals (28.33%) from the northern section, 21 animals (35%) from the central section, and 22 animals (36.67%) from the southern section.”
Point 30. Lines 175 – 178 Change to read “Interaction with fishing gear could be identified in 40% (N = 24) of carcasses. Of turtles that interacted with fishing gear, 62.5% (N = 15) were found to have interacted with longline gear,whereas 37.5% (N = 9) were found to have interacted with a fishing net.”
Response 30. The text has been corrected as suggested.
L 199–202 “Interaction with fishing gear could be identified in 40% (N = 24) of carcasses. Of turtles that interacted with fishing gear, 62.5% (N = 15) were found to have interacted with longline gear, whereas 37.5% (N = 9) were found to have interacted with a fishing net.”
Point 31.Lines 178 – 179 and Lines 181 – 182 seem to contradict each other.
Specific pathological findings which indicated interaction with longline gear were drowning, a deeply placedfishing hook, and/or monofilament lines through the gastrointestinal tract with consequent fibrino-necrotic enteritis. Autopsy findings of drowning … strongly suggested that entanglement in trawl gear was the cause of death.
It appears that drowning was used to assign cause of death to both longline and trawl gear. Please clarify.
Response 31. Authors’ comment: Drowning is an immediate cause of death resulting from interaction with longline gear, as indicated by the presence of fishing hooks and monofilament line. The absence of specific lesions, along with the signs of drowning, suggests entanglement in trawl gear. In order to avoid contradiction, the text has been changed as suggested.
L 202–204 “Specific pathological findings which indicated interaction with longline gear were drowning, a deeply placed fishing hook, and/or monofilament lines through the gastrointestinal tract with consequent fibrino-necrotic enteritis (Figure 4, A).
Point 32.Lines 207 – 211 Please provide SE for mean sizes presented.
Response 32. The authors have decided to provide CI instead of SE.
L 241–247 “Longline fishing gear was found to be a major cause of death in small juvenile turtles (CCL mean size = 41.4 cm, CI 29.11–53.69 cm, range = 20–100 cm), while colliding with a vessel was the prevailing cause of death in larger juvenile and adult turtles (CCL mean size = 64.11 cm, CI 54.42–73.81 cm, range = 35–100 cm). Trawl nets (CCL mean size = 58.33 cm; CI 41.32–75.35 cm, range 30–100 cm) and cold stunning affect a wide range of loggerhead turtle sizes (CCL mean size = 55.75 cm, CI 41.86–69.64 cm, range = 18–70 cm).”
Point 32. Line 211 Authors state that “Differences found in the determined cause of death among size class categories were significant (p < 0.05).” So, there were significant differences between all size classes and all causes of death? There was a significant difference between small juveniles and adults that died of Interstitial pneumonia (5.26% and 5%, respectively)? Need to address concerns about the impact of such a small sample size.
Response 33. Many thanks to the reviewer for spotting potentially false statistical interference due to the small sample size. It's better to admit that the data is not sufficiently conclusive to make any inference. The concerns have been addressed.
L 247–249 “Due to such a small sample size, the differences found in the determined cause of death among size class categories were not sufficiently conclusive to make any inference.”
Point 34. Lines 213 – 216 This paragraph seems out of place. No other discussion of stomach contents. While interesting, it doesn’t add to the paper. Remove it.
Response 34. The authors agree with comments that it doesn’t add to the paper, but, as suggested by another reviewer, the sentence has been rewritten with more information added.
L 250–252 “Irregular fragments of non-transparent black plastic caps, ranging from 1 to 3 cm in size, were found in the gastrointestinal tracts of two subadult animals, causing ulceration and haemorrhagic lesions with partial obstruction of the intestines.”
DISCUSSION
Point 35. Overall it seems like this section was pieced together and some statements are not supported by the results, or contradict the results presented. Need to make sure that information in discussion is not just repetitive of background information. Several incomplete sentences and ideas. Need to clearly connect conclusions to results. Would be good to see references to previous studies that are supported by, or contradict, the results of this study. It is hinted at, but the connection to previous studies in not fully made.
Response 35. The authors acknowledge the reviewer's comments on our discussion section. They recognize the potential bias in data obtained from turtle strandings, as previously noted in the Introduction and Discussion section. Consequently, the connections to previous studies may not be as apparent.
Point 36. Lines 220 – 22 The sentence “The analysis demonstrated clear seasonality of strandings along the Croatian Adriatic coast, which is indicative of season-specific abundance in different areas and season-specific or age-specific mortality” needs to be clarified.
There is support for season-specific mortality, but not age-specific mortality related to seasonality (as stated in the Results). Additionally, only mention
Response 36. The text has been changed as suggested.
L 259–262 “The analysis demonstrated clear seasonality of strandings along the Croatian Adriatic coast, which is indicative of season-specific abundance in different areas and age-specific mortality related to region, which was observed among small juveniles.”
Point 37. Paragraph of Lines 251 – 266 Authors are presenting new results and appear to reference information not previously presented in Figure 3. Need to move Results to the Results section. How do the 4 high density areas related to the three parts in the study?
Response 37. The text has been rewritten as suggested.
L 165–169 “In the northeast Adriatic, the kernel density analysis identified two geographically divided, high-density stranding areas during the summer months, one along the Istrian peninsula and the other along the coast of Lošinj Island (Figure 3, Points 1 and 2). Two additional high-density stranding areas, identified in the Zadar and Šibenik regions, are located in the central Adriatic (Figure 3, Points 3 and 4).”
L 292–297 “The results suggest that, in the northeast Adriatic, high-density stranding areas along the Istrian peninsula and the coast of Lošinj Island might be significant during the summer months. Turtle rescue centres were established in both areas, one in Pula and the other in Mali Lošinj. The results for the two additional key areas, Zadar and Šibenik (Figure 3, Points 3 and 4), located in the central Adriatic, indicated that the majority of strandings occurred during January and August.”
Point 38. Lines 285 – 286 Authors state “it was determined post-mortem that 40% of fishing-induced mortality indicates interaction with fishing gear as a major cause of overall turtle stranding occurrences”
Do you mean “it was determined post-mortem that 40% of mortality was associated with interaction with fishing gear, suggesting that by-catch is a major cause of turtle strandings along the Croatian Adriatic Coast.”
Response 38. The text has been corrected as suggested.
L 327–329 “Nevertheless, it was determined post-mortem that 40% of mortality was associated with interaction with fishing gear, suggesting that by-catch is a major cause of turtle strandings along the Croatian Adriatic coast.”
Point 39. Is there any information related to seasonal or area fishing effort that could be compared to the stranding data?
Response 39. That information wasn’t available to the authors. Public information about fishing vessels (with fishing gear) for 2023 does exist (as part of a scientific project by the Institute of Oceanography and Fisheries and the Ministry of Agriculture, Directorate of Fisheries). Legislation requires reporting catches and fishing zones for each vessel. By combining those datasets, it would be possible to correctly estimate seasonal and area fishing efforts and compare it to stranding data. We would need to assume that the fishing effort has not changed over 13-year period.
Point 40. Did you look at causes of mortality in relationship to the three sections? How might the distribution of necropsied turtles bias your research.
Response 40. Many thanks to the reviewer for pointing out potential bias in our research. Additional analysis has been performed to assess whether the necropsied turtles were a representative sample of strandings among sections?
|
Stranding location |
|
Necropsy |
||
|
|
Performed |
Not performed |
Total |
|
|
Northern |
17 (5.96) |
268 (94.04) |
285 (100) |
|
|
Central |
21 (10.14) |
186 (89.86) |
41 (36.94) |
|
|
Southern |
22 (17.19) |
106 (82.81) |
26 (23.42) |
|
|
Total |
60 (9.68) |
560 (90.32) |
111 (100) |
|
The authors have decided to change the text without presenting the table.
“The cause of death did not differ significantly among areas (p = 0.411) and months (p = 0.125).” – the sentence has been deleted.
Some parts of the text have been rewritten.
L 228–233 “The differences between sections in conducted versus non-conducted turtle necropsies were notably significant (p = 0.004). Only 17 (5.96%) stranded animals from the northern section, 21 (10.14%) from the central section, and 22 (17.19%) from the southern section were subjected to a necropsy. Consequently, necropsy findings cannot indicate potential differences in the cause of mortality between sections. “
L 309–313 “In our study, the majority of strandings were recorded on beaches near towns and tourist centres, which could indicate reporting bias towards easier detection near populated areas along the Croatian coast. This could also be a potential reason for the observed discrepancy in the number of stranded animals submitted to a necropsy among different areas.”

Reviewer 4 Report
Comments and Suggestions for Authors
This research is interesting, since it focuses on both the spatiotemporal distribution and the main threats to its conservation. The ms in general terms is well written, with a concrete and punctual vision of the question to be answered. However, in the attached document, some observations and suggestions for the authors are included.
On the other hand, due to the number of threats that Caretta caretta presents to this region, the authors must include conservation plans and proposals in their conclusion. Consider the possibility of negotiating with the authorities of your country the implementation of public policies to regulate certain activities that put species at risk, among other actions.

A minor revision is necessary
Author Response
Authors’ Response to Reviewer 4
Point 1. L84-85 detail the sampling method. Did they draw a transect, quadrant? Were they only based on opportunistic sampling? Was there a systematic sampling? Did they carry out night sampling to record arrivals during the nesting season?
Response 1 To describe the sampling method in the analysed dataset we have rephrased L 94–99.
„The passive surveillance data presented in this study was collected in the Croatian Adriatic Sea during a 13-year period, spanning from January 2010 to December 2022, as part of the Protocol for Alerting and Monitoring of dead, sick or injured strictly protected marine species (marine mammals, sea turtles, and cartilaginous fish) mandated by the Croatian Ministry of Environment and Energy.”
Authors' comment: The passive surveillance of strandings is clearly opportunistic sampling (without a sampling frame) — any sea turtle found alive or dead on the beaches or floating in coastal waters was considered stranded, and reported to Croatian Ministry of Environment and Energy, with several constraints addressed in manuscript (L 306–312).
Point 2. L88 indicate units (SI) in which the record was taken (cm? mm?)
Response 2. Correction: L 92 (CCL in cm).
Point 3. L145 -149 I suggest that these results be presented in a table, comparing months and sites.
Response 3. Many thanks to the reviewer for pointing out the visuality of spatiotemporal analysis. Authors preferred presenting the data in Figure 2 format rather than a 13 x 4 Table. The text and Figure 2 have been corrected.
L152 “The spatiotemporal distribution of stranding records is presented in Figure 2. and statistical analysis showed statistically significant differences in monthly records between sections (p = 0.001).”
L172 *Numbers in columns correspond to the number of records in a certain month and section
Point 4. L205 “I suggest that the authors use a photograph to illustrate each case. “
Response 4. The authors have decided to add Figure 4, and have made corresponding changes in the text and figure captions.
Figure 4. Gross appearance of interaction with longline gear, trauma caused by vessel collisions, drowning after cold stunning, and interstitial pneumonia
*A – Jejunum, stomach, and intestine, gross appearance of interaction with longline gear; B – Dorsal side of the body, trauma of the carapace caused by vessel collisions; C – Ventral side of the body, cyanosis, and skin haemorrhage on the neck and flipper; D – Lungs, interstitial pneumonia
L 201–203 “Specific pathological findings which indicated interaction with longline gear were drowning, a deeply placed fishing hook, and/or monofilament lines through the gastrointestinal tract with consequent fibrino-necrotic enteritis (Figure 4, A).”
L 211–214 “Pathological findings included extensive sharp-edged traumatic lesions of the head, neck, carapace or plastron with corresponding trauma and haemorrhages in deeper tissues (Figure 4, B).”
L 222–224 “The turtles were diagnosed with cachexia and showed minimal pathologic signs that included froth in the airways, cyanosis and skin haemorrhage on the neck and flippers, or interstitial pneumonia (present in treated animals, Figure 4, C).”
L 225–226 “Interstitial pneumonia was diagnosed in three (5%) animals, which died in rescue centres after treatment (Figure 4, D).”
Point 5. On the other hand, due to the number of threats that Caretta caretta presents to this region, the authors must include conservation plans and proposals in their conclusion. Consider the possibility of negotiating with the authorities of your country the implementation of public policies to regulate certain activities that put species at risk, among other actions.
Response 5. A new sentence with additional information suggested by reviewer 2 has been added to the Conclusion.
L 375–379 “One of the targeted conservation efforts we propose to authorities is the establishment of mobile treatment facilities that could be a quick response measure to found debilitated and injured sea turtle (and other endangered and protected species), and could provide first aid and secure transport to rescue centres.”

Round 2
Reviewer 3 Report
Comments and Suggestions for Authors
Authors have addressed the major concerns and justified the use of CI in the results.